# A Second-Order Method for Removing Mixed Noise from Remote Sensing Images

**DOI:** 10.3390/s23177543

**Published:** 2023-08-30

**Authors:** Ying Zhou, Chao Ren, Shengguo Zhang, Xiaoqin Xue, Yuanyuan Liu, Jiakai Lu, Cong Ding

**Affiliations:** 1College of Geomatics and Geoinformation, Guilin University of Technology, Guilin 541006, China; zhouying01@glut.edu.cn (Y.Z.); 2120222038@glut.edu.cn (X.X.); 2120222009@glut.edu.cn (J.L.); dc3050367@gmail.com (C.D.); 2Guangxi Key Laboratory of Spatial Information and Geomatics, Guilin 541106, China; 3PowerChina Guiyang Engineering Corporation Limited, Guiyang 550081, China; 18177393593@163.com

**Keywords:** remote sensing image, mixed noise, DnCNN, adaptive median filtering, nearest neighbor pixel weighted median

## Abstract

Remote sensing image denoising is of great significance for the subsequent use and research of images. Gaussian noise and salt-and-pepper noise are prevalent noises in images. Contemporary denoising algorithms often exhibit limitations when addressing such mixed noise scenarios, manifesting in suboptimal denoising outcomes and the potential blurring of image edges subsequent to the denoising process. To address the above problems, a second-order removal method for mixed noise in remote sensing images was proposed. In the first stage of the method, dilated convolution was introduced into the DnCNN (denoising convolutional neural network) network framework to increase the receptive field of the network, so that more feature information could be extracted from remote sensing images. Meanwhile, a DropoutLayer was introduced after the deep convolution layer to build the noise reduction model to prevent the network from overfitting and to simplify the training difficulty, and then the model was used to perform the preliminary noise reduction on the images. To further improve the image quality of the preliminary denoising results, effectively remove the salt-and-pepper noise in the mixed noise, and preserve more image edge details and texture features, the proposed method employed a second stage on the basis of adaptive median filtering. In this second stage, the median value in the original filter window median was replaced by the nearest neighbor pixel weighted median, so that the preliminary noise reduction result was subjected to secondary processing, and the final denoising result of the mixed noise of the remote sensing image was obtained. In order to verify the feasibility and effectiveness of the algorithm, the remote sensing image denoising experiments and denoised image edge detection experiments were carried out in this paper. When the experimental results are analyzed through subjective visual assessment, images denoised using the proposed method exhibit clearer and more natural details, and they effectively retain edge and texture features. In terms of objective evaluation, the performance of different denoising algorithms is compared using metrics such as mean square error (MSE), peak signal-to-noise ratio (PSNR), and mean structural similarity index (MSSIM). The experimental outcomes indicate that the proposed method for denoising mixed noise in remote sensing images outperforms traditional denoising techniques, achieving a clearer image restoration effect.

## 1. Introduction

Remote sensing facilitates the acquisition of valuable information pertaining to objects and regions from a considerable distance, employing active methodologies such as radar and lidar, as well as passive approaches like multispectral and hyperspectral techniques [1]. As a result of environmental conditions, transmission channels, and various contributing factors, images invariably encounter noise during the stages of acquisition, compression, and transmission. This noise intrusion subsequently leads to the distortion and degradation of vital image information. The existence of such noise significantly hampers subsequent image-related tasks, including, but not limited to, video processing, image analysis, and tracking [2]. Image denoising is a classical and continually evolving subject within the realm of low-level computer vision, focusing on the restoration of a high-quality image from its degraded counterpart. This domain remains notably vibrant, embodying an essential cornerstone in various practical contexts like digital photography, medical image analysis, remote sensing, surveillance, and digital entertainment [3]. Consequently, the paramount goal in the realm of remote sensing image preprocessing has consistently revolved around augmenting image quality by adeptly eliminating noise, all the while safeguarding the inherent edge details and textural attributes of the original image to the greatest extent feasible.

Currently, the processing of remote sensing images primarily focuses on Gaussian noise and salt-and-pepper noise [4,5]. In recent times, a plethora of denoising methodologies and models have been introduced, aiming to mitigate noise interference and elevate the overall quality of images. Chang et al. [6] combine a total variation model with sparse representation for denoising remote sensing images. Yan et al. [7] designed a remote sensing image denoising algorithm based on two-dimensional empirical mode decomposition and quaternion wavelet transform. However, the denoised images still lack sufficient clarity, with varying degrees of distortion in image details. Xu et al. [8] propose a denoising method for satellite remote sensing images by integrating principal component analysis and complex wavelet transform. This method first extracts features using noise-adjusted principal component analysis and then applies complex wavelet transform to denoise the low-energy components. Xia et al. [9] apply K-SVD (K-singular value decomposition) sparse representation theory to denoise satellite remote sensing images. Zhang et al. [10] address the non-local self-similarity and sparsity in remote sensing images and propose a sparse denoising algorithm based on non-local self-similarity. Dabov et al. [11] introduce the BM3D (block-matching and 3D filtering) algorithm, which exploits the correlation between image blocks and utilizes joint 3D filtering to achieve image denoising. While these approaches proficiently address Gaussian noise in images, they exhibit limited efficacy when it comes to eliminating salt-and-pepper noise. Moreover, due to the intricacies inherent in the optimization challenges, they frequently demand substantial time and computational resources to achieve operational efficiency. Thanh D N H et al. [12] propose an adaptive total variation (TV) regularization model for salt-and-pepper denoising in digital images. However, for high-density noise, some details are lost. With the booming development of deep learning technology, Zhang et al. [13] propose the DnCNN (denoising convolutional neural networks) model for forward denoising, which achieves promising results in Gaussian denoising for traditional natural images. Nevertheless, challenges persist in denoising remote sensing images characterized by intricate textures, as the denoising outcomes often tend to manifest in the form of blurred edge textures. Wu et al. [14] utilize a residual autoencoder network combined with edge enhancement for remote sensing image denoising. Although this method effectively removes Gaussian noise from images, it also struggles with salt-and-pepper noise removal. Wang et al. [15] propose a new method to denoise UAV (unmanned aerial vehicle) images, which introduces a novel deep neural network method based on generative adversarial learning to trace the mapping relationship between noisy and clean images. The denoised images generated by this proposed method enjoy clearer ground object edges and more detailed textures of ground objects. Due to the image processed by the denoising method based on convolutional neural network exhibiting a fuzzy phenomenon in texture details, Khan A et al. [16] propose and apply a generative adversarial network-based image denoising training architecture to multiple-level Gaussian image denoising tasks. This innovative framework effectively addresses the blurriness issue by prompting the denoiser network to learn the distribution of sharp, noise-free images, thus avoiding the production of blurry outcomes. It is important to note, however, that the scope of this study is exclusively limited to addressing additive white Gaussian noise. Khmag A’s research [17] employs a self-adjusting generative adversarial network (GAN). The proposed method combines noise suppression and an adaptive learning GAN procedure in order to guarantee the removal of unwanted additive white Gaussian noise. But the time complexity of this approach needs to be reduced.

Nonetheless, in real-world scenarios, noise within remote sensing images frequently comprises a composite of Gaussian and salt-and-pepper noise. Conventional denoising techniques alone exhibit suboptimal performance when confronted with the intricacies of remote sensing images imbued with such blended noise patterns. Therefore, Zhu et al. [18] propose a three-layer combined filtering method that integrated Bayes wavelet threshold filtering, adaptive Wiener filtering, and adaptive median filtering to remove mixed noise. Li et al. [19] propose an image denoising method for mixed noise based on quaternion non-local low-rank and total variation. This method effectively denoises and suppresses artifacts while better preserving image details and color information. Zheng et al. [20] establish a bridge between the factor-based regularization and the HSI (hyperspectral image) priors and propose a double-factor-regularized LRTF (low-rank tensor factorization) model for HSI mixed noise removal. Deng et al. [21] present a combined filtering denoising method that combines the three-dimensional block matching algorithm with adaptive median filtering. Zhao et al. [22] combine the effective noise reduction capability of the BM3D algorithm for Gaussian noise and propose an integrated BM3D method for removing mixed noise from remote sensing images. Ren et al. [23] propose a denoising method that combines BM3D with multilevel nonlinear weighted average median filtering to remove mixed noise in remote sensing images. Despite the efficacy of amalgamated filtering techniques in achieving mixed noise reduction, residual noise may persist due to the interplay between filters of varying dimensions.

Therefore, this study proposes a two-stage denoising method for remote sensing images with mixed noise, combining deep learning and spatial domain filtering. In the first stage, an extension convolution is introduced based on the DnCNN denoising model to increase the network’s receptive field, enabling the extraction of more feature information in complex remote sensing images. Additionally, a DropoutLayer is incorporated after the deep convolutional layers to prevent overfitting, simplify network training, and facilitate the denoising process using aerial photography as data-driven training for the denoising model. The trained model is then applied to perform initial denoising on the images. In the second stage, based on adaptive median filtering, a weighted median replacement of the nearest neighboring pixel within the filtering window is employed to process the initially denoised results. This aims to improve the image quality of the initial denoised results, effectively remove salt-and-pepper noise from the mixed noise, and preserve more detailed image edges and texture features.

The rest of this paper is organized as follows: Section 2 briefly introduces the materials and methods used for the DnCNN denoising model, expansion convolution, the DropoutLayer, and adaptive median filtering; Section 3 describes the algorithm constructed in this article; Section 4 introduces the experimental setup, and results are given and discussed; and finally, Section 5 offers the conclusion and final remarks on the paper.

## 2. Materials and Methods

### 2.1. DnCNN Denoising Model

Based on a convolutional neural network (CNN), the DnCNN model uses a series of methods such as residual learning, regularization, and batch normalization to improve the denoising performance of the model, which can effectively remove Gaussian noise contained in the image. The DnCNN algorithm network architecture is mainly composed of three parts. The first part is the first layer, composed of Conv + ReLU; Conv is a convolution kernel [24] of size 3 × 3 × C; there are a total of 64; the step size is 1 × 1; C is used to distinguish grayscale images and color images; if the input image is a grayscale map, then C = 1; if the input image is a color map, then C = 3; “ReLU” stands for rectified linear unit, a popular activation function commonly used in neural networks [25]. It introduces non-linearity by outputting the input directly if it is positive; otherwise, it outputs zero. This activation function has been proven effective in enhancing the learning capabilities of neural networks. The second part is layers 2 to (d − 1; each layer is composed of Conv + BN + ReLU; Conv is of a size of 3 × 3 × 64 convolution kernels; the number of convolution kernels in each layer is 64; the step size is 1 × 1; BN [26] is the batch normalization layer of 64 channels; and ReLU is the activation function. The third part, the final layer, consists of Conv, which uses C 3 × 3 × 64 filters to reconstruct the output of the processed image.

### 2.2. Expansion Convolution

Broadening the receptive field of a network is a standard approach to gather enhanced contextual information within convolutional neural networks. Currently, prevalent methods for extending the receptive field primarily involve augmenting network depth, increasing filter size, and employing expansion convolution techniques. However, increasing the network depth will lead to the degradation of network performance, and expanding the filter size will introduce more parameter numbers and increase the computation of the network, while the expansion convolution can expand the receptive field without increasing the amount of network computation [27]. The principle of dilated convolution is to inject holes into the standard convolution kernel to increase the receptive field of the network, so it is also called void convolution or dilated convolution. For example, for an expansion convolution with a convolution kernel size of 3 × 3, a dilation factor of 2, a step size of 1, and a number of layers of n, the network receptive field size can be expressed as (4n + 1) × (4n + 1); for an ordinary convolution with a convolution kernel size of 3 × 3, a step size of 1, and a number of layers of n, the network receptive field size is expressed as (2n + 1) × (2n + 1).

### 2.3. DropoutLayer

DropoutLayer is a method proposed by Hinton et al. [28] that can improve the generalization ability of network models and solve the problem of network overfitting in deep learning. The principle is to set a constraint on the weight of each implied unit that obeys the Bernoulli distribution, and if this constraint is activated, the unit will be temporarily discarded from the network with P probability so as to discard some features, improve the generalization ability of the network, and achieve the purpose of solving network overfitting [29].

### 2.4. Adaptive Median Filtering

Traditional adaptive median filtering (AMF) is a nonlinear filter that not only effectively removes noise, but also preserves edge texture detail to some extent. This method is mainly divided into two processes in the denoising process, which can be defined as process A and process B. Let X(i,j) be the window corresponding to the central pixel (i,j) when filtering (the maximum allowable size of the window is $M_{max}$); $Z_{min}$ is the minimum value of the gray value in window X(i,j); $Z_{max}$ is the maximum value of the gray value in window X(i,j); $I_{med}$ is the median value of the gray value in window X(i,j); and Z(i,j) is the gray value at the position of the image pixel (i,j). Let
(1)$$Z_{A1}=I_{med}-Z_{min}$$
(2)$$Z_{A2}=Z_{max}-I_{med}$$
(3)$$Z_{B1}=Z_{\left(i,j\right)}-Z_{min}$$
(4)$$Z_{B2}=Z_{max}-Z_{\left(i,j\right)}$$

The specific algorithm flow of filtering is as follows:(1)Process A: if $Z_{A1}>0$ and $Z_{A2}>0$, go to Process B; otherwise, increase the size of window X(i,j). If the window size is equal to or less than the maximum window size $M_{max}$, process A is repeated; otherwise, the grayscale value Z(i,j) for that pixel is output;(2)Process B: if $Z_{B1}>0$ and $Z_{B2}>0$, output the grayscale value Z(i,j) of the pixel; otherwise, output the median value $I_{med}$.


## 3. The Algorithm Constructed in This Article

While the DnCNN noise reduction model yields commendable outcomes in mitigating Gaussian noise in natural images, its denoising efficacy often falters when applied to remote sensing images marked by intricate topographical attributes. This is particularly true for images encompassing mixed noise patterns, where the denoising outcomes might retain a higher degree of noise. Furthermore, the preservation of edge details and texture information tends to be less distinct in such cases. Therefore, based on the DnCNN model and adaptive median filtering, a second-order method for mixing noise from remote sensing images is proposed.

In the first stage, the first-order noise reduction model DP-DnCNN is built with the DnCNN network structure as the basic framework; the network depth is set to 20 layers, and in the even layers of layers 1–19, the expansion factor of 2 is used to increase the receptive field of the network and improve the feature extraction ability of remote sensing images, and the odd layers and the 20th layer use ordinary convolution (Conv). A DropoutLayer with a probability of 0.5 is added after the ReLU layer of layer 19 to prevent overfitting of the network and improve the generalization ability of the model. The network structure is mainly divided into five parts: the first part is the input layer, which is composed of an ImageinputLayer; the second part is layer 1, consisting of Conv + ReLU; the third part, which is composed of Conv + BN + ReLU and DilatedConv + BN + ReLU and Conv + BN + ReLU + DropoutLayer, is the 2nd–19th layers; the fourth part is the 20th layer, consisting of a Conv; the fifth part is the regression output layer, which consists of a RegressionLayer. The input layer of the network input has an image size of 50 × 50 × 1, the input channel of layer 1 is 1, the convolution kernel size is 3 × 3, and the output channel is 64. The input channel of layers 2–19 is 64, the convolution kernel is 3 × 3, and the output channel is 64. The input channel of layer 20 is 64, the convolution kernel size is 3 × 3, and the output channel is 1.

In the second stage, since the traditional adaptive median filtering is replacing the noise pixel with the median value within its filtering window, the size of the median value in the window will directly affect the denoising effect and image clarity, and when the median pixel is far away from the noise point to be replaced, it will lead to image distortion and blurring. Therefore, this paper improves the traditional adaptive median filtering and proposes adaptive nearest neighbor weight median filtering (RW-IAMF). The specific improvement method is as follows: for the median output of process B, the nearest neighbor pixel weighted median value is used to replace the value in the original filter window, so that the median value of the output pixel is closer to the original image element, which improves the denoising performance of the algorithm and the retention ability of edge details and texture features. Figure 1 below is a schematic diagram of the nearest neighbor pixels, in which the four pixels of Z(i−1,j), Z(i,j−1), Z(i+1,j), and Z(i,j+1) represent the nearest neighbor pixels of point Z(i,j). 

Firstly, the closest neighbor pixel value set $W[f\left(i,j\right)]$ of the noise is selected in the current filter window, and then the noise detection judgment is carried out on the selected nearest neighbor pixel value: if $f_{n}(i,j)$ = 0 or 255, $f_{n}(i,j)$ is judged as a noisy pixel and removed, where $f_{n}(i,j)$ is a pixel value in the closest neighbor pixel set; the median value $Med(W[f\left(i,j\right)]$ is taken from the set of pixels of the nearest neighborhood after detection, and then based on this median, the weighting calculation method from reference [30] is used to calculate the weighting coefficient of each pixel in the set by applying Formulas (5) and (6), and then the remaining pixels in the set of nearest neighborhood pixels after detection are weighted and summed with the corresponding weighting coefficients obtained by using Equation (7), and finally, the calculation results are used as filter output.
(5)$$sum=\sum_{n=1}^{N}\left(1÷(1+{\left(f_{n}\left(i,j\right)-Med(W[f(i,j)])\right)}^{2})\right)$$
(6)$$w_{n}\left(i,j\right)=\frac{1÷(1+{\left(f_{n}\left(i,j\right)-Med(W[f(i,j)])\right)}^{2})}{sum}$$
(7)$$f\left(i,j\right)=\sum_{n=1}^{N}f_{n}(i,j)\times w_{n}(i,j)$$

Among these, $Med\{W\left[f\left(i,j\right)\right]\}$ is the set of pixel values of the nearest neighborhood, $f_{n}(i,j)$ is a pixel value in the closest neighbor pixel set, N is the total number of pixels remaining in the closest neighbor pixel set $W[f\left(i,j\right)]$ after detection, $w_{n}(i,j)$ is the weighting coefficient size of a pixel obtained, and f(i,j) is the filter output result.

The proposed algorithm combines the noise reduction model trained in the first stage with the improved adaptive median filtering in the second stage, which effectively removes the mixed noise while improving the protection ability of image edge details and texture features. Firstly, the first-stage noise reduction model DP-DnCNN is used to initially reduce the noise of the images containing mixed noise, and then RW-IAMF is used to make secondary corrections to the output images to improve the ability of the algorithm to remove salt-and-pepper noise from mixed noise. The specific algorithm framework flow is shown in Figure 2 below.

## 4. Experimental Setup and Results Analysis

### 4.1. Network Model Training

The training of the DP-DnCNN network model in this paper utilized the ITCVD aerial imagery dataset provided by the University of Twente Research Information as the training set. The dataset includes 135 aerial images with dimensions of 5616 × 3744 × 3 pixels and a resolution of 0.1 m. To facilitate training, Gaussian noise with a mean of 0 and a variance ranging from 0.001 to 0.01 was randomly added to the images. Furthermore, each image was randomly cropped into 512 small sub-blocks of size 50 × 50, for a total of 69,120 small sub-blocks that were used for training. The input size for network training was set to 50 × 50 pixels; the network training solver used the stochastic gradient descent with momentum (SGDM) optimizer; the initial learning rate was set to 0.01, and it decreased by a factor of 1/10 every 10 epochs; and the size of the number of mini-batch image blocks was set to 128. To stabilize the training process and prevent gradient explosion, gradient clipping was employed, with a specified gradient threshold to 0.005. The gradient thresholding method used considered the absolute value of the gradients. The L2 regularization factor parameter was set to 0.0001 to reduce network overfitting. Network training was based on MATLAB programming, while the hardware configuration included an Intel(R) Core(TM) i7-11700 @2.50 GHz pro-cessor and an Nvidia GeForce RTX 3060 graphics card for acceleration. The memory size was 12 GB, and the operating system employed was Windows 10.3.2.3.

### 4.2. Evaluation Metrics

The evaluation of denoising effectiveness employs mean square error [31] (MSE), peak signal-to-noise ratios [32] (PSNR), mean structural similarity [33] (MSSIM), and edge detection results as assessment metrics for the algorithm’s precision in removing mixed noise from images. In these metrics, a smaller MSE and a larger PSNR indicate better denoising quality, while MSSIM is closer to human visual evaluation, with higher values signifying more intact post-denoising image structures.

The calculation formula for MSE is as follows:(8)$$MSE=\frac{\sum_{i=1}^{M}\sum_{j=1}^{N}{\left[F_{1}\left(i,j\right)-F_{2}\left(i,j\right)\right]}^{2}}{M\times N}$$

Here, $F_{1}(i,j)$ and $F_{2}(i,j)$ represent the original and denoised images, respectively, and M and N denote the image dimensions.

The PSNR calculation formula is
(9)$$PSNR=10\times \log_{10}(255^{2}÷MSE)$$

The MSSIM calculation formula involves a structure similarity index (SSIM) computed over image blocks and then averaged to yield MSSIM. It is defined as
(10)$$MSSIM=\frac{1}{B}\underset{i=1}{\overset{B}{\sum }}{SSIM}_{i}$$
(11)$$SSIM=\frac{\left(2\mu_{x}\mu_{y}+c_{1})(2\sigma_{xy}+c_{2}\right)}{\left({\mu_{x}}^{2}+{\mu_{y}}^{2}+c_{1})({\sigma_{x}}^{2}+{\sigma_{y}}^{2}+c_{2}\right)}$$
where SSIM represents structural similarity; x and y denote the original and denoised images, respectively;$\;\mu_{x}$ is the mean of x and $\mu_{y}$ is the mean of y;$\;\sigma_{x}$ is the standard deviation of x and $\sigma_{y}$ is the standard deviation of y;$\;\sigma_{xy}$ is the covariance of x and y;$\;c_{1}={\left(k_{1}L\right)}^{2}$ and $c_{2}={\left(k_{2}L\right)}^{2}$ are constants to ensure stability; B represents the number of image blocks.

These metrics collectively provide a comprehensive assessment of the denoising performance of the algorithm and its ability to effectively remove mixed noise from remote sensing images.

### 4.3. Remote Sensing Image Denoising Experiment

To validate the feasibility of the proposed algorithm, this paper conducted remote sensing image denoising experiments using imagery captured by the Gaofen-2 satellite. The image data were cropped to a size of 400 × 400 pixels. Three different concentrations of noise were added to the intercepted images: 0.001/0.003, 0.003/0.005, and 0.005/0.008, where the former value was the Gaussian noise variance with a mean of 0, and the latter was the noise density of salt-and-pepper noise. The variance of zero-mean Gaussian noise characterizes the amplitude of random fluctuations introduced to pixel values within the image. The magnitude of variability in pixel values due to zero-mean Gaussian noise is quantified by the variance. The variance captures the extent of randomness incorporated into the image pixel values through zero-mean Gaussian noise. The density of salt-and-pepper noise signifies the frequency of occurrence of isolated bright (“salt”) and dark (“pepper”) pixels across the image. The proportion of isolated bright and dark pixels within the image, referred to as salt-and-pepper noise density, illustrates the prevalence of this noise type. The abundance of isolated bright and dark pixel pairs within the image, indicated by salt-and-pepper noise density, elucidates the spatial distribution of this noise phenomenon. This allowed for quantitative and qualitative analysis of the denoising results. The experiment employed mean square error (MSE), peak signal-to-noise ratio (PSNR), and mean structural similarity (MSSIM) as evaluation metrics for denoising performance. The proposed algorithm in this paper (DPRW) was compared with the IAMF algorithm, the BM3D algorithm, the DnCNN algorithm, DnCNN combined with adaptive median filtering (DNIA), DP-DnCNN combined with adaptive median filtering (DPIA), DnCNN combined with the RW-IAMF (DNRW) algorithm, method from reference [21] (BMIA), and method from reference [23] (BMDJ), in terms of denoising effectiveness. The experimental results are shown in Table 1, Table 2 and Table 3. A smaller MSE value and a larger PSNR value indicate better denoising quality, while a higher MSSIM value signifies a more complete restoration of image structures, preserved edge details, and texture information, which is closer to human visual perception. By comparing the highlighted data with other data in Table 1, Table 2 and Table 3, it can be observed that for mixed noise in remote sensing images, the proposed algorithm in this paper achieves a lower MSE and a higher PSNR compared to other algorithms, indicating its superior denoising performance. In terms of the MSSIM metric, the proposed algorithm also outperforms the other comparative algorithms, demonstrating that it preserves more complete image structures, edge details, and texture features after denoising, compared to other denoising algorithms.

In addition to the qualitative and quantitative evaluation using the MSE, PSNR, and MSSIM metrics, the denoising effectiveness of each algorithm is also compared at the visual level in this study. Figure 3 presents the denoising results of each algorithm under a mixed noise concentration of 0.003/0.005. Analyzing Figure 3 reveals that the proposed algorithm not only effectively removes the mixed noise present in the images but also preserves the image structures, edge details, and texture features more comprehensively, resulting in clearer images.

Direct visual comparison may not be easily discernible; therefore, it is preferable to zoom in on specific local regions and annotate them with red bounding boxes to facilitate a clear observation of the effects. Further analysis of the localized magnified images of the denoising results for each algorithm (Figure 4) indicates that, compared to the combined algorithms, the individual DnCNN and BM3D algorithms can effectively remove Gaussian noise from the images but perform poorly in denoising the images containing salt-and-pepper noise, resulting in significant residual noise. Additionally, the BM3D algorithm exhibits edge detail loss and blurring in the texture areas in the smooth regions of the images. The IAMF algorithm performs well in denoising salt-and-pepper noise but exhibits poor denoising effectiveness for Gaussian noise. The DNIA, BMIA, and DNRW algorithms can effectively remove the mixed noise from the images, but there are still some residual noise points. Compared to DNIA and BMIA algorithms, the DNRW algorithm has relatively fewer residual noise points due to the influence of the weighted median of neighboring pixels. The BMDJ algorithm can more thoroughly remove the mixed noise from the images but suffers from edge detail loss in the denoised results and introduces a certain level of blurring in the texture-smooth regions. The DPIA algorithm yields clearer denoised images with more preserved image structures compared to the previous algorithms, but it still has some residual noise points.

Comparing the denoising results of all the aforementioned algorithms, the proposed algorithm in this paper (DPRW) consistently outperforms the other methods, yielding denoised results that are visually closer to the real images. This approach not only proficiently eliminates mixed noise, yielding images of heightened clarity, but also retains a greater quantity of edge details and textural information, thereby culminating in more comprehensive image structures.

### 4.4. Image Edge Detection

To provide a more intuitive illustration of the proposed algorithm’s ability to preserve edge details in images, the Canny operator is used to perform edge detection on the denoised results of each algorithm. Edge detection primarily focuses on extracting information about the boundaries of objects in an image, making it a commonly used image extraction method in the field of computer vision [34]. Edge extraction is essentially a form of filtering, and the accuracy of the Canny operator is relatively ideal within the entire edge detection algorithm. The results of edge detection using the Canny operator on the image are shown in Figure 5. From the figure, it can be observed that compared to the original image and the detection results of the proposed algorithm, the single IAMF algorithm produces numerous cluttered and discontinuous lines in its detection results. This indicates that the image denoised using this method still contains some residual noise. Comparing the edge detection results of the DnCNN algorithm with the original image, it can be observed that there are some scattered and irregular contour lines. This indicates that, due to the presence of salt-and-pepper noise, the denoised images still retain a significant amount of noise. Furthermore, from the figure, it can be seen that certain details in the image are also mistakenly filtered out as noise. In the edge detection results of the BM3D algorithm, there is a noticeable loss of certain details and discontinuities in the contour lines. As for the DPIA, DNRW, and DNIA algorithms, their edge detection results after denoising still exhibit some discontinuous lines and a partial loss of object contours compared to the original image and the edge detection results of the algorithm proposed in this paper. This indicates that these three algorithms still contain some residual noise after denoising and fail to preserve the edge details of the image effectively. Regarding the BMIA and BMDJ algorithms, many crucial edges are missed, and the detected edges show poor continuity with noticeable breaks. This indicates that during the denoising process, some edge details from the original image are treated as noise and filtered out, leading to the inability to preserve the edge details of the image effectively.

Contrasted with the previously discussed methodologies, the denoised images obtained through the proposed algorithm showcase smoother and more continuous edge contours, closely resembling the edge detection outcomes derived from the original image. This observation underscores the dual effectiveness of the proposed algorithm: not only does it proficiently eliminate mixed noise from the images, but it also retains a greater portion of image edge details. This preservation holds significant implications for downstream image classification and recognition applications.

## 5. Conclusions

The presence of noise in remote sensing imagery can significantly impact its subsequent use and analysis, making denoising an essential step in remote sensing image processing. To address the issue of mixed noise in the imagery, this study employs aerial photography as training data to drive the process. In the first stage of our proposed method, we enhance the Gaussian denoising model based on the foundational DnCNN architecture by incorporating dilated convolution and a DropoutLayer. This augmentation empowers the method’s capability for Gaussian noise reduction. Additionally, in the second stage, we synergize this enhanced model with an adaptive nearest neighbor weighted median filtering. This harmonious fusion results in a novel dual-stage approach adept at effectively mitigating the intricate issue of mixed noise in remote sensing imagery.

In an analysis of the results of remote sensing image denoising experiments and edge detection on denoised images, the proposed method outperforms other algorithms in comparison. It effectively addresses challenges that other methods struggle with, achieving higher peak signal-to-noise ratios (PSNR), improved mean structural similarity (MSSIM), and lower mean square error (MSE). For instance, at a noise density of 0.003/0.005, the MSE is reduced by at least approximately 2%, the PSNR increases by up to about 3 dB, and the MSSIM improves by up to approximately 40%. The edge detection results also demonstrate that while removing noise, the proposed algorithm preserves more of the original image’s detail and edge contour information, thus achieving remarkable denoising outcomes. This bears significant practical implications for the subsequent use of images. Therefore, whether viewed subjectively or assessed objectively for quality, the method proposed in this paper exhibits commendable denoising performance, outperforming traditional methods. It effectively eliminates Gaussian and salt-and-pepper mixed noise, adeptly retains edge details and texture features, resulting in clearer image outcomes, and holds potential for real-world applications in actual remote sensing image denoising processes.

## Figures and Tables

**Figure 1 sensors-23-07543-f001:**
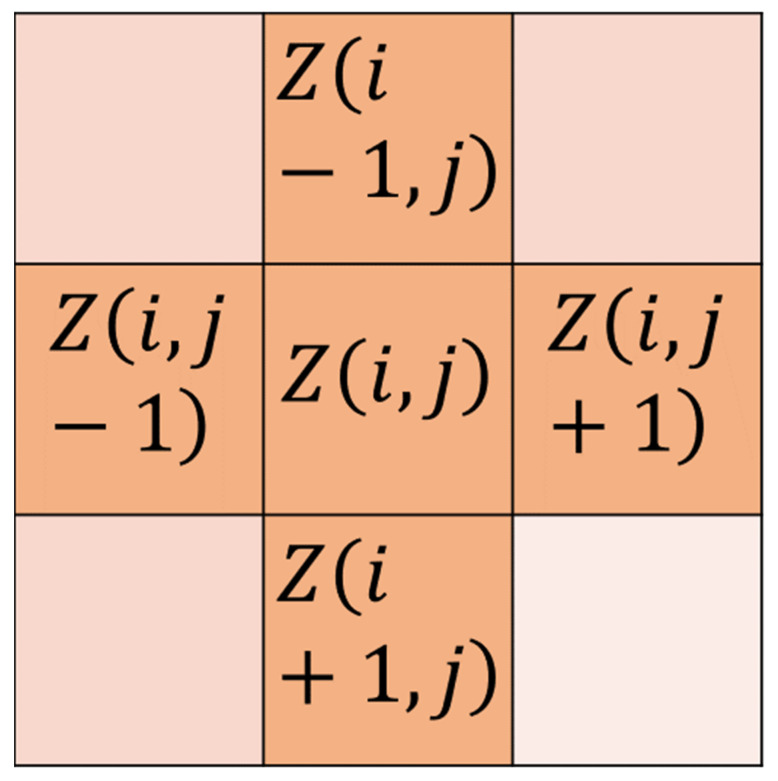
Schematic diagram of the nearest neighbor pixel.

**Figure 2 sensors-23-07543-f002:**
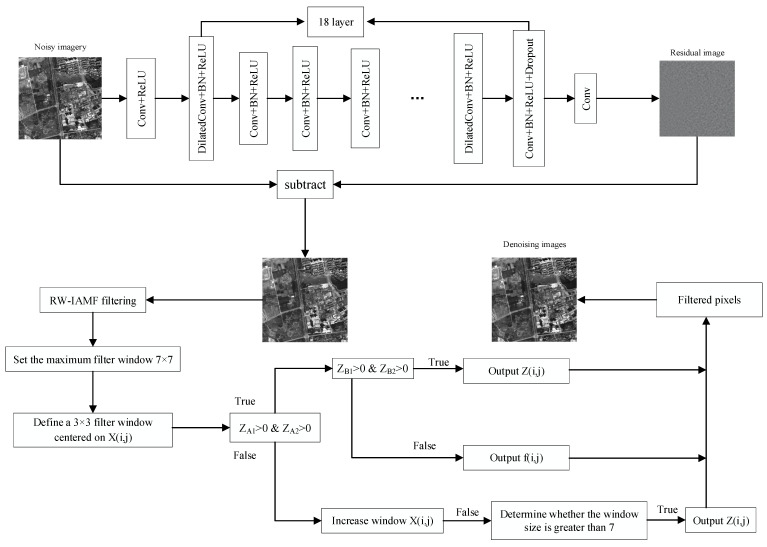
Flowchart of the algorithm framework of this paper.

**Figure 3 sensors-23-07543-f003:**
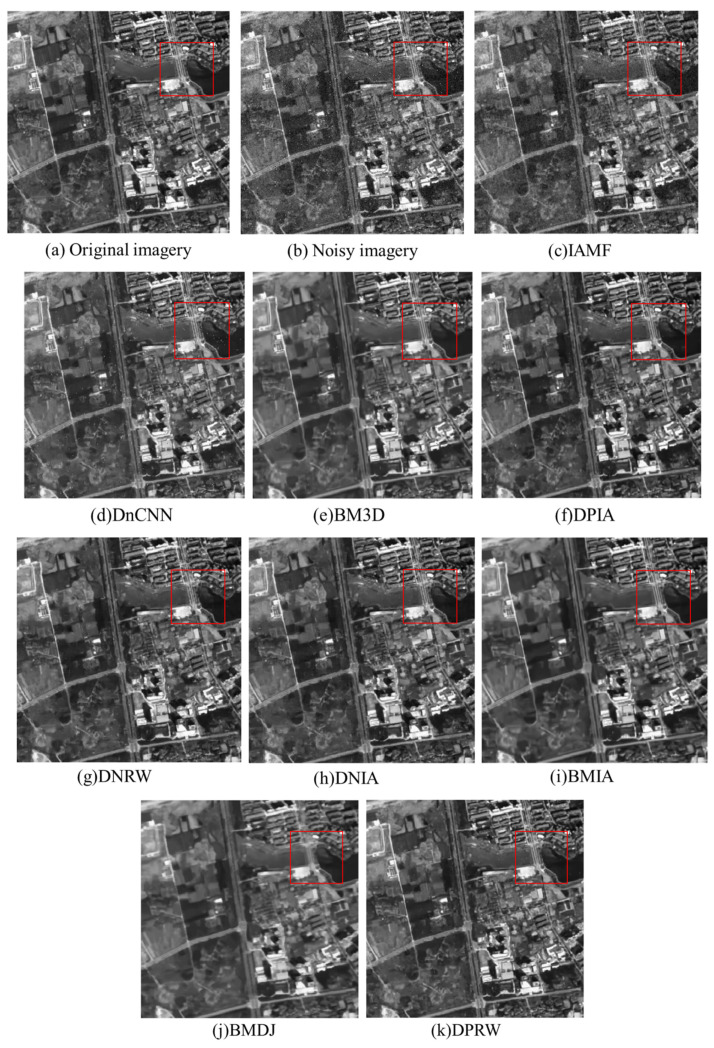
Denoising effect diagram of each algorithm. Due to direct visual comparison may not be easily discernible; therefore, it is preferable to zoom in on specific local regions and annotate them with red bounding boxes to facilitate a clear observation of the effects.

**Figure 4 sensors-23-07543-f004:**
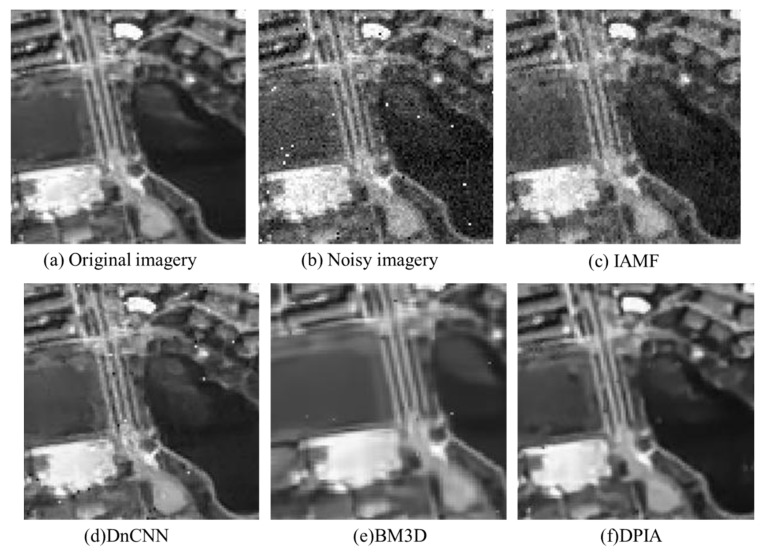
Partial enlarged view of denoising effect of each algorithm. The areas highlighted with red boxes in Figure 3 are magnified locally. (**a**) shows the magnified view of the original image; (**b**) shows the magnified view of the noisy image; (**c**–**k**) correspond to the magnified views of IAMF, DnCNN, BM3D, DPIA, DNRW, DNIA, BMIA, BMDJ. (**k**) represents the magnified view of the image after denoising with the proposed algorithm in this study (DPRW).

**Figure 5 sensors-23-07543-f005:**
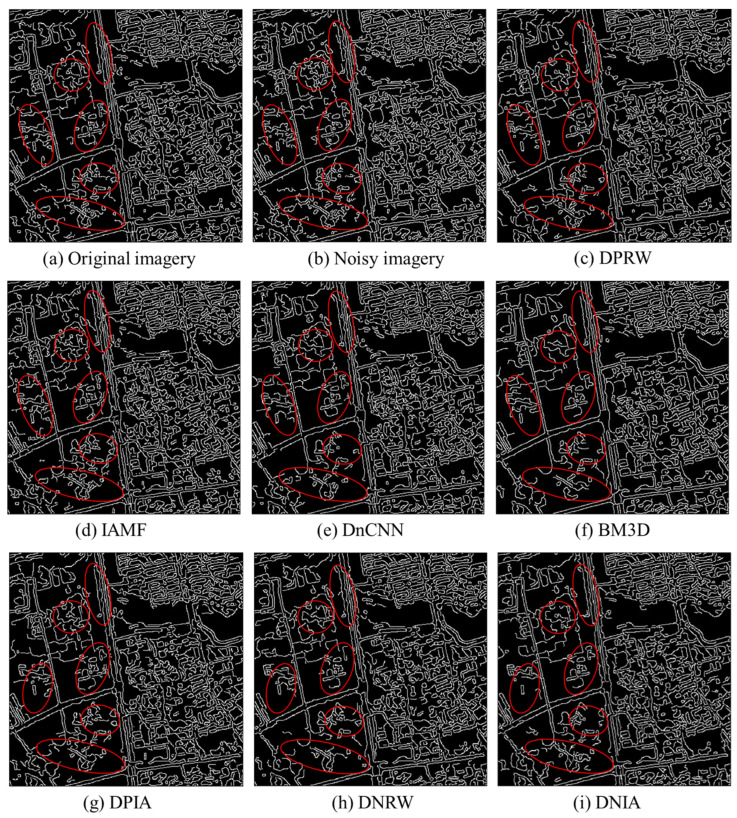
Image edge detection results of denoising images by using each algorithm. The areas highlighted with red circles are regions of particular significance for comparing the edge detection results among different algorithms.

**Table 1 sensors-23-07543-t001:** Comparison of mean square error (MSE) of image denoising results by different methods.

Noise Concentration	IAMF	BM3D	DnCNN	DNIA	DPIA	DNRW	BMIA	BMDJ	DPRW
0.001/0.003	26.0992	26.9251	33.7315	18.2286	17.6437	18.3868	27.8081	39.1686	**17.3953**
0.003/0.005	50.8343	29.8920	39.2509	28.3363	28.0865	28.4099	30.1804	40.4189	**27.8908**
0.005/0.008	63.6574	33.0005	42.6203	34.9392	33.8542	34.3924	33.6824	42.7185	**33.5601**

**Table 2 sensors-23-07543-t002:** Comparison of peak signal-to-noise ratio (PSNR) of image denoising results by different methods.

Noise Concentration	IAMF	BM3D	DnCNN	DNIA	DPIA	DNRW	BMIA	BMDJ	DPRW
0.001/0.003	33.9645	33.8292	32.8505	35.5233	35.6649	35.4857	33.6891	32.2014	**35.7265**
0.003/0.005	31.0692	33.3752	32.1923	33.6074	33.6464	33.5961	33.3336	32.0624	**33.6762**
0.005/0.008	30.0923	32.9456	31.8346	32.6977	32.8347	32.7662	32.8568	31.8246	**32.8726**

**Table 3 sensors-23-07543-t003:** Comparison of the mean structural similarity (MSSIM) of the image denoising results of each method.

Noise Concentration	IAMF	BM3D	DnCNN	DNIA	DPIA	DNRW	BMIA	BMDJ	DPRW
0.001/0.003	0.8145	0.8322	0.7878	0.8865	0.8908	0.9570	0.8285	0.7451	**0.9580**
0.003/0.005	0.6616	0.8213	0.7513	0.8319	0.8357	0.9241	0.8215	0.7401	**0.9264**
0.005/0.008	0.5708	0.8039	0.7208	0.7913	0.7991	0.8989	0.8100	0.7294	**0.9036**

## Data Availability

The dataset used in this work are available online with open access for non-commercial research use only. The ITCVD aerial imagery dataset is available online at https://research.utwente.nl/en/datasets/itcvd-dataset (accessed on 28 August 2023).

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
