# Peer review of "A Second-Order Method for Removing Mixed Noise from Remote Sensing Images"

_sensors, 2023, doi:10.3390/s23177543_

Round 1

Reviewer 1 Report

The paper is well presented and it can be accepted for publication with some English editing. 

Good, and it needs some modifications 

Reviewer 2 Report

This paper performed to A Second-order Method for Removing Mixed Noise from Remote Sensing Images. The article is, in general, well written but there are some issues that authors should consider to revise in order to improve its quality. Some comments were done in this way:

·         Abstract should be expanded sentences related to the results. The results of the study should be given as numerical percentages.

·         Revise Figure 1. Make it clearer and higher resolution.

·         Also give the experimental results with graphs. Also give with a table.

·         Let's fix grammatical errors throughout the article.

·         The article should be edited completely according to the journal writing guide.

·         Fractions should be given with dots throughout the article, including figures and tables.

·         Conclusions should be written in more detail adding numeric data.

Reviewer 3 Report

This paper proposes a two-stage denoising method for remote sensing images with mixed noise, combining deep learning and spatial domain filtering.

The references are not enough and important references are missing.

The analysis of the reason that the proposed method is better for the mixed noise is not clear. The proposed method should aim at the characters of the mixed noise.

Moderate editing of English language required

Reviewer 4 Report

Dear authors, thank you for submitting of your article.

comments:

Abstract

DnCNN network - by first use any abbreviation it is necessary to define it

what should be "DropoutLayer layer"?

Introduction:

Remote sensing is a technology that collects information about the Earth in a non-contact way [1].

- here should be a citation of the  remote sensing definition, from the "Manual of  Remote sensing" for example, not [1], it is about denoising, wrong reference

the same [2], you write about basic information on remorte sensing, why reference on image fusion?-

row 80 BM3D method - define it

row 169 what is ReLu

This part is incomprehensible, rewrite it or use a flowchart with explanations of abbreviations:

The network structure is mainly divided into five parts, the first part is the input layer, which is composed of an ImageinputLayer; The second part is layer 1, consisting of Conv + ReLU; The third part is the 2~19th layer, which is composed of Conv + BN + ReLU and DilatedConv + BN + ReLU and Conv + BN + ReLU + DropoutLayer; The fourth part is the 20th layer, consisting of a Conv; The fifth part is the regression output layer, which consists of a RegressionLayer. The input layer of the network input has an image size of 50×50×1, the input channel of layer 1 is 1, the convolution kernel size is 3×3, and the output channel is 64. The input channel of layers 2~19 is 64, the convolution kernel is 3×3, and the output channel is 64. The input channel of layer 20 is 64, the convolution kernel size is 3×3, and the output channel is 1.

row 246 Three different concentrations of noise were added to the intercepted images: 0.001/0.003, 0.003/0.005, 0.005/0.008

- it's unclear to the reader

The text is full of abbreviations; some are not explained, others are, they repeat themselves. Perhaps you'd better give an overview of the abbreviations (see table 1 for example)

why do you use artificially added noise in your images and not real noisy images? E.g. typical pepper and salt images are radar images.

why do you use artificially added noise in your images and not real noisy images? E.g. typical pepper and salt images are radar images

Conclusion:

this study employs aerial photography as training data- aerial ?

The conclusion is too general and needs to be improved and completed.
In general, the article does not bring too much new knowledge, it is more suitable for a conference paper.

References:

-use the template, what is [J]? journal? Improve these, doi  - if exist- will be usefull

-from 24 references only 4 are outside China; I think noise in remotely sensed data, this is being dealt with worldwide

Round 2

Reviewer 3 Report

No more comments

No more comments

Reviewer 4 Report

Dear authors, thank you for your improved article.

I already wrote in the first reading:Figure 3 is unnecessary, just one as an overview. There are normally no differences in the individual sub-images.

fig.5 also here it was better to give details that would show the differences, as in fig.4

I suggest improving Figures 3 and 5. There is no visible difference.

Otherwise, the article is ok. Thank you.
